# Significant Serpents: Predictive Modelling of Bioclimatic Venom Variation in Russell's Viper

**Navaneel Sarangi**◉, **R. R. Senji Laxme**◉, **Kartik Sunagar** ◉ *

Centre for Ecological Sciences, Indian Institute of Science Bangalore, India

◉ These authors contributed equally to this work.
* ksunagar@iisc.ac.in

## Abstract

### Background

Russell's viper (*Daboia russelii*) is the clinically most important snake species in the world. Considerable variation has been documented in *D. russelii* venoms across the Indian subcontinent, which can drive the diverse envenomation profiles in snakebite victims. Therefore, understanding the role of ecological and environmental factors influencing the compositional and functional variation can provide critical insights into the complex evolutionary adaptations of this species and pave the way for the development of targeted therapies.

### Methods

We examined the influence of bioclimatic factors on *D. russelii* venom functions by analysing 115 samples sourced from various locations across India. The enzymatic activities of major toxins, such as proteases and phospholipases, were estimated to capture the functional variation in these venoms. Multiple regression models were developed to evaluate the relationship between venom variability and the historical climate data, specifically temperature and precipitation. Furthermore, predictive models were employed to map venom phenotypes across the distribution range of *D. russelii*.

### Findings

Our findings reveal a collective influence of various temperature and precipitation parameters that partly explain the variability in enzymatic activities of *D. russelii* venom. Our models effectively captured regional differences in venom composition and linked climatic conditions with functional variations.

### Conclusion

This study highlights the significant role of abiotic factors in shaping the venom profiles of Russell's vipers across India. The predictive venom phenotype maps developed from our models can guide the deployment of targeted therapies and treatment protocols across the biogeographically diverse Indian subcontinent and improve clinical treatment

**Data availability statement:** The dataset used in the analysis, and those generated from the analysis, are included in the supplementary information of the manuscript. The R codes used in the study are included in the supplementary information of the manuscript.

**Funding:** KS was supported by the Wellcome Trust DBT India Alliance Fellowship (IA/I/19/2/504647). RRSL was supported by the Prime Minister's Research Fellowship (PMRF) from the Ministry of Human Resource Development (MHRD). The funders had no role in study design, data collection and analysis, decision to publish, or preparation of the manuscript.

**Competing interests:** The authors have declared that no competing interests exist.

outcomes of *D. russelii* envenoming. This research enhances our understanding of venom phenotype evolution and has practical implications for improving snakebite management.

## Author summary

Russell's viper (*Daboia russelii*), with a near-country-wide distribution across the Indian subcontinent, causes the highest number of snakebite deaths and disabilities. A considerable geographic venom variation documented in *D. russelii*, among others, makes it the clinically most relevant snake species. However, biotic and abiotic factors that drive this variation remain uninvestigated. Hence, we assessed the influence of bioclimatic factors on the functional venom variation through statistical modelling. *D. russelii* venoms (n=115) were sourced from varied biogeographies across >6600 km, followed by the functional characterisation of major toxins. Multiple regression models generated revealed the impact of historical climatic conditions on venom variability. Phenotype maps that predict venom biochemistries of *D. russelii* across its range distribution were generated, providing valuable insights for deploying targeted therapies. Overall, our findings provide novel insights into the effects of abiotic factors on snake venom variation and its implications for enhancing the effectiveness of snakebite therapeutics.

## 1. Introduction

Organismal adaptations are shaped by fundamental interactions between biotic and abiotic components of an ecosystem. Snake venom, a key innovation underpinning the evolutionary success of snakes on land and in water, is one such example. As a highly adaptive trait, snake venoms are theorised to evolve under various biotic and abiotic selective pressures [1]. Prey availability, predator pressure, temperature, elevation, humidity, and precipitation, among other things, may collectively play a role in determining the composition and potency of snake venoms [2,3]. Moreover, being ectothermic, fluctuations in bioclimatic conditions may directly affect the physiology of snakes and, in turn, the functional characteristics of their venoms [4]. While several ecological, environmental, evolutionary and genetic factors have been theorised to shape this intraspecific variation [2,5–9], only a handful of studies have attempted to delineate the correlation between predictors and venom composition and function [10–12]. A deeper understanding of the drivers of venom variation could provide fascinating insights into the origin and diversification of key evolutionary innovations.

India, a biogeographically diverse landmass where the environmental conditions vary considerably across short distances [13], is infamous as the snakebite capital of the world. Russell's Viper (*Daboia russelii*), which accounts for over 43% of snakebite-related mortalities annually [14], results in various pathological symptoms across its near-country-wide distribution, causing devastating life and economic loss. Most victims are young males from rural agrarian communities, who are the primary breadwinners of their families and major contributors to India's agricultural economy [15]. The intraspecific variation in the venom composition of *D. russelii* is well recognised [16–19]. While we are beginning to understand the influence of the biotic factors in driving this variation, how abiotic factors shape *D. russelii* venoms remains uninvestigated.

Therefore, in this study, we integrated historical environmental data with functional venom variation and employed statistical modelling to analyse the role of bioclimatic factors in shaping *D. russelii* venoms across India. Our findings reveal that a combination of temperature and precipitation collectively explains the variation observed in the enzymatic activities

of the major venom toxins of *D. russelii*. Moreover, using the models generated in this study, we constructed a predictive map of patterns of venom variation across the range distribution of this species. Since these toxins constitute a large proportion of Russell's viper venom and are responsible for clinically severe symptoms, we can predict *D. russelii* venom compositions, functions and clinical manifestations across India. Such predictive 'venom phenotype maps' may facilitate the effective deployment of targeted snakebite therapeutics in future. Thus, we highlight the significant role of bioclimatic factors in shaping snake venoms and the crucial application of predictive modelling in managing snakebites.

## 2. Materials and methods

### 2.1 Ethical statement

No animal or human samples have been used for this study. All venoms were collected with appropriate forest department permission from the relevant State Forest Departments: Punjab - PB (3615/11/10/12), Tamil Nadu - TN (WL5(A)/33005/2017, Permission No. 61/2023) and (C.No.WL1/32767/2021-3(1) dated 03.11.2021), Andhra Pradesh - AP (13526/2017/WL-3), West Bengal - WB (386/WL/4R-6/2017) and (1023/WL/4R-12/2022), Rajasthan - RJ (P.3(3) Forest/2004), Maharashtra - MH (Desk-22(8)/Research/CR-80(16–17)/943/2017-18) and (22(8)/WL/Research/CR-60(17-18)/3349/21-22), Goa - GA (2-66-WL-Research Permissions-FD-2022-23-Vol.IV/858), Karnataka - KA (PCCF(WL)/E2/CR-06/2018-19) and PCCF(W1)/ C1(C3)/CR-09/2017-18, Kerala - KL (KFDHQ-1006/2021-CWW/WL10) and Madhya Pradesh - MP (TK-1/48-II/606).

### 2.2 Sampling and protein estimation

Venom samples were collected from 115 individuals of adult *D. russelii*, individually or by pooling, across India (Fig 1). Each sample was snap-frozen using liquid nitrogen and subsequently lyophilised and stored at -80 °C until further analysis (Table A in S1 Text). Protein concentrations of crude venoms were quantified using a modified Bradford assay in triplicates (Table A in S1 Text; [20]). The absorbance was measured at 595 nm on an EPOCH2 microplate reader (BioTek, USA), where Bovine Serum Albumin (BSA) served as a standard.

### 2.3 PLA$_2$ assay

In the PLA$_2$ assay, we mixed 5 µg protein of each venom sample with 100 µl of 500 mM NOB substrate (4-nitro-3-(octanoyloxy) benzoic acid) (Enzo Life Sciences, New York, USA). The total volume of the reaction mixture was adjusted to 120 µl using the reaction buffer (10 mM Tris-HCl, 10 mM CaCl2, and 100 mM NaCl at a pH of 7.8) and incubated at 37 °C for 40 minutes. To monitor the reaction kinetics, absorbance was measured at 425 nm every 10 minutes using an EPOCH2 microplate spectrophotometer (BioTek, USA). A standard curve was generated by parallelly incubating various concentrations of the NOB substrate with 4 N NaOH. The PLA$_2$ activity, indicated by the rate of NOB substrate hydrolysis, was quantified in nanomoles per minute per milligram of venom by extrapolating from this standard curve [21,22]. The assay was performed in triplicates, and one-way ANOVA was performed to compare intrapopulation variability in the samples using GraphPad PRISM 8 (GraphPad Software, San Diego, California USA, www.graphpad.com).

### 2.4 Protease assay

For the analysis of proteolytic activity, 10 µg of venom protein was incubated with azocasein substrate at 37 °C for 90 minutes. To terminate the reaction, 200 µl of trichloroacetic acid

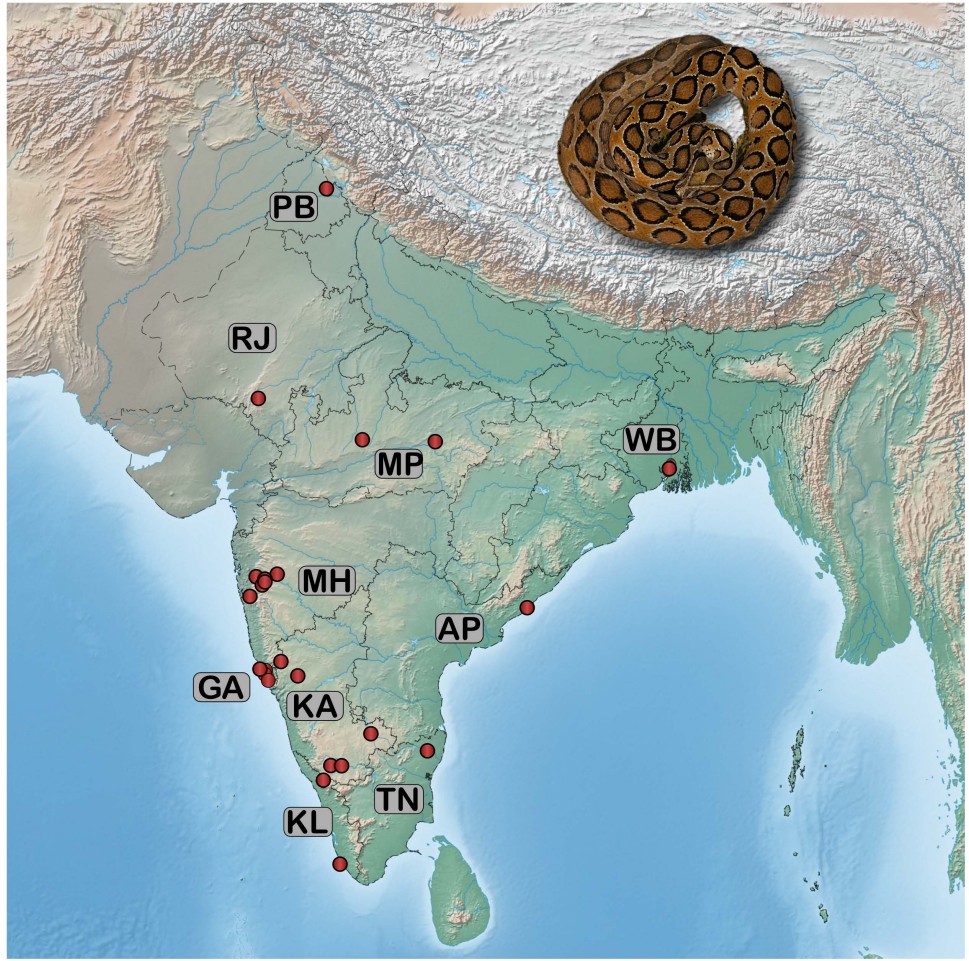

**Fig 1. Sampling locations.** Point locations from where the samples were collected are depicted on a map of India. The samples were collected from distinct biogeographic zones. The map of India shown here was prepared with QGIS 3.8 [19] (Table A in S1 Text).

was added, followed by centrifugation at 1000 x g for 5 minutes to separate the supernatant, which was then neutralised with an equivalent volume of 0.5 M NaOH. The absorbance of the reaction mixture was recorded at 440 nm using an EPOCH2 microplate spectrophotometer (BioTek, USA). Purified bovine pancreatic protease (Sigma-Aldrich, USA) served as a standard for comparison [23,24]. The assay was performed in triplicates, and one-way ANOVA was performed to compare intrapopulation variability in the samples using GraphPad PRISM 8 (GraphPad Software, San Diego, California USA, www.graphpad.com).

## 2.5 LAAO assay

For estimating the activity of LAAO, 0.5 µg of venom protein was prewarmed at 37 °C for 10 minutes. This step was followed by the addition of 200 µL reaction mixture (5 mM L-leucine, 50 mM Tris-HCl buffer, 5 IU/ml horseradish peroxidase, and 2 mM o-phenylenediamine dihydrochloride) and incubation at 37 °C for an additional 10 minutes. The enzymatic reaction was halted by adding 50 µL 2 M $H_2SO_4$ solution. Absorbance was then measured at 492 nm in an EPOCH2 microplate spectrophotometer (BioTek, USA). To quantify the enzymatic activity, we generated a standard curve using different concentrations of $H_2O_2$ and

the reaction mixture. The cleavage rate of the L-leucine substrate, expressed in nanomoles per minute per milligram of venom, was deduced from this standard curve [22,25]. The assay was performed in triplicates, and one-way ANOVA was performed to compare intrapopulation variability in the samples using GraphPad PRISM 8 (GraphPad Software, San Diego, California USA, www.graphpad.com).

## 2.6  Bioclimatic variable selection and data extraction

Bioclimatic variables relevant to our study were selected based on a list of 14 meteorological features detailed by the United States Geological Association (USGA) [26]. These climatic indices are available as Geographic Information System (GIS) raster surfaces for climate data from 1895 to 2009. These bioclimatic predictors were used to understand the relationship between abiotic factors and the distribution of venom functions, as described previously [11,12]. From this comprehensive list, variables were carefully chosen to align with our sampling strategy. For example, seasonal factors, such as Precipitation of Wettest Month, Max Temperature of Warmest Month, and Mean Temperature of Driest Quarter, were excluded as the venom activity of the same individual from different seasons was unavailable. The selection process aimed to capture a broad yet relevant range of bioclimatic factors that influence enzymatic activity. The variables that provide local and global insights, along with monthly and annual data on variations in temperature and precipitation, were downselected. GeoTIFF files containing bioclimatic data were downloaded from the WorldClim database (http://www.worldclim.org, accessed on 14.04.2023). These were further processed in R using the Raster package [27,28]. GeoTIFF files were clipped, and the vector data encompassing the venom sampling locations were extracted.

## 2.7  Linear regression model development

The independent influence of bioclimatic variables in predicting the enzymatic activities of *D. russelii* venoms was determined by performing a Simple Linear Regression (SLR) analysis in R (R version 4.2.2, 2022-10-31). The significance of the model was tested using the p-values generated by a T-test on the model parameter estimates. The prediction accuracy was inferred from the adjusted coefficient of determination ($R^2$). Logarithmic, inverse and square root transformations were performed on the dependent variables to correct for normality and heteroscedasticity violations of the dataset.

Further, a Multiple Linear Regression (MLR) model was constructed to understand the combined effect of multiple bioclimatic variables on enzymatic activities. Following the above-mentioned strategy, the MLR models were built using the untransformed and transformed datasets. These models were refined through stepwise regression to eliminate non-significant variables using t-statistics, thereby optimising the adjusted $R^2$ values. The significance of the obtained models was assessed from the p-values of the F-test. Additionally, the Akaike Information Criterion (AIC) has been reported to provide robust statistical support to the omodel. The normality (Wilk-shapiro Test), homoscedasticity (Breusch-Pagan), linearity (Rainbow Test), and multicollinearity [variance inflation factor (VIF)] tests were performed to corroborate the reliability of the generated SLR and MLR models. Geographically weighted regression was also performed to account for potential biases stemming from spatial autocorrelation of enzymatic data (Table B in S1 Text). The analysis failed to identify any potential issues due to spatial autocorrelation. The R script for the analyses is provided in fileS1 Text. The following R packages were used for model development, verification and validation: psych [29], lm.beta [30], lmtest [31], sandwich [32], car [33], tidyverse [34], DHARMa [35], MASS [36], olsrr [37], jtools [38] and moments [39].

## 2.8  Spatial data acquisition and predictive mapping

We optimised the significant MLR models for each enzymatic activity and used historical bioclimatic data as model input to predict enzymatic activities across the Indian subcontinent. The variables were downloaded individually as GeoTIFF files at 1 km² resolution (http://www.worldclim.org, accessed on 14.04.2023). Each shapefile was cropped to a minimum and maximum extension of latitude and longitude (68.1 E to 97.4 E and 6.74 N to 35.7 N). The R packages (R version 4.2.2, 2022-10-31) used for this prediction mapping included rgdal [40], raster [27], terra [41], rasterVis [42] and sf [43]. The elaborate codes are provided in file S1 Text. These predictions were then mapped as a gradient on the map of India.

## 3.  Results

### 3.1  Functional characterisation of venoms

To quantify the functional variations arising from compositional differences, various biochemical assays were performed on the venoms collected from the pan-Indian populations of *D. russelii* (Fig 1). In the PLA₂ assay, substantial inter- and intrapopulation variability was observed (Fig 2A). Notably, venoms from distinct locations across eastern India (WB) exhibited stark intrapopulation variability in the $PLA_2$ activities, with values ranging between 38 – 315.2 nmol/mg/min. This was followed by western India (MH), where $PLA_2$ activities ranged between 16.3 – 138.4 nmol/mg/min. Venoms from northern (PB) and southern (TN) India exhibited activities above 100 nmol/mg/min, while samples from all other locations, including western (RJ), southeastern (AP), central (MP) and the other southern regions (KA and KL) of India exhibited activities ranging between 14.6 – 95.9 nmol/mg/min. The lowest activities were observed for samples from southwestern (GA) India: 4.9 - 44.9 nmol/mg/min.

Similarly, a broad range of values was recorded for the relative proteolytic activities across the sampled populations (Fig 2B). While most samples from various populations exhibited between 29 – 40% relative proteolysis, certain populations had venoms with little to no proteolytic activity. For instance, maximum intrapopulation variation was observed for samples from western India (WB), followed by one of the southern Indian (KA) populations. When we assessed the variation in LAAO activity of Russell's viper venoms, the results revealed notable intra- and interpopulation variability across the sampled regions of India (Fig 2C). The activity levels ranged between 131 – 16472 nmol/mg/min within one of the southern Indian (KA) populations. A similar intrapopulation variation was observed within the southwestern (GA) population, with LAAO activities ranging between 483 - 9627 nmol/mg/min. A range of intermediate LAAO activities were found across other regions: central (MP), western (MH), eastern (WB), southern (TN, KL) and southeastern (AP) populations: 45.7 – 10763 nmol/mg/min (Fig 2C). Lower LAAO activities were observed for northern (PB) and western (RJ) Indian regions (918 - 1729 nmol/mg/min).

### 3.2  Simple Linear Regression (SLR) models

SLR was performed to delineate the influence of bioclimatic factors (Table 1) on functional activities of the venoms of the pan-Indian populations of Russell's viper (Table C in S1 Text). The model identified Precipitation Seasonality (PS) as a significant predictor of $PLA_2$ activity (p = 0.00876 and $R^2$ = 0.178). After transforming the data to account for heteroscedasticity and normality, the SLR model posits that for every unit increase in PS, there is an associated 0.56 unit increase in the $PLA_2$ activity if all the other variables were constant. Notably, this model surpasses the 99% confidence threshold of prediction accuracy for the given sample

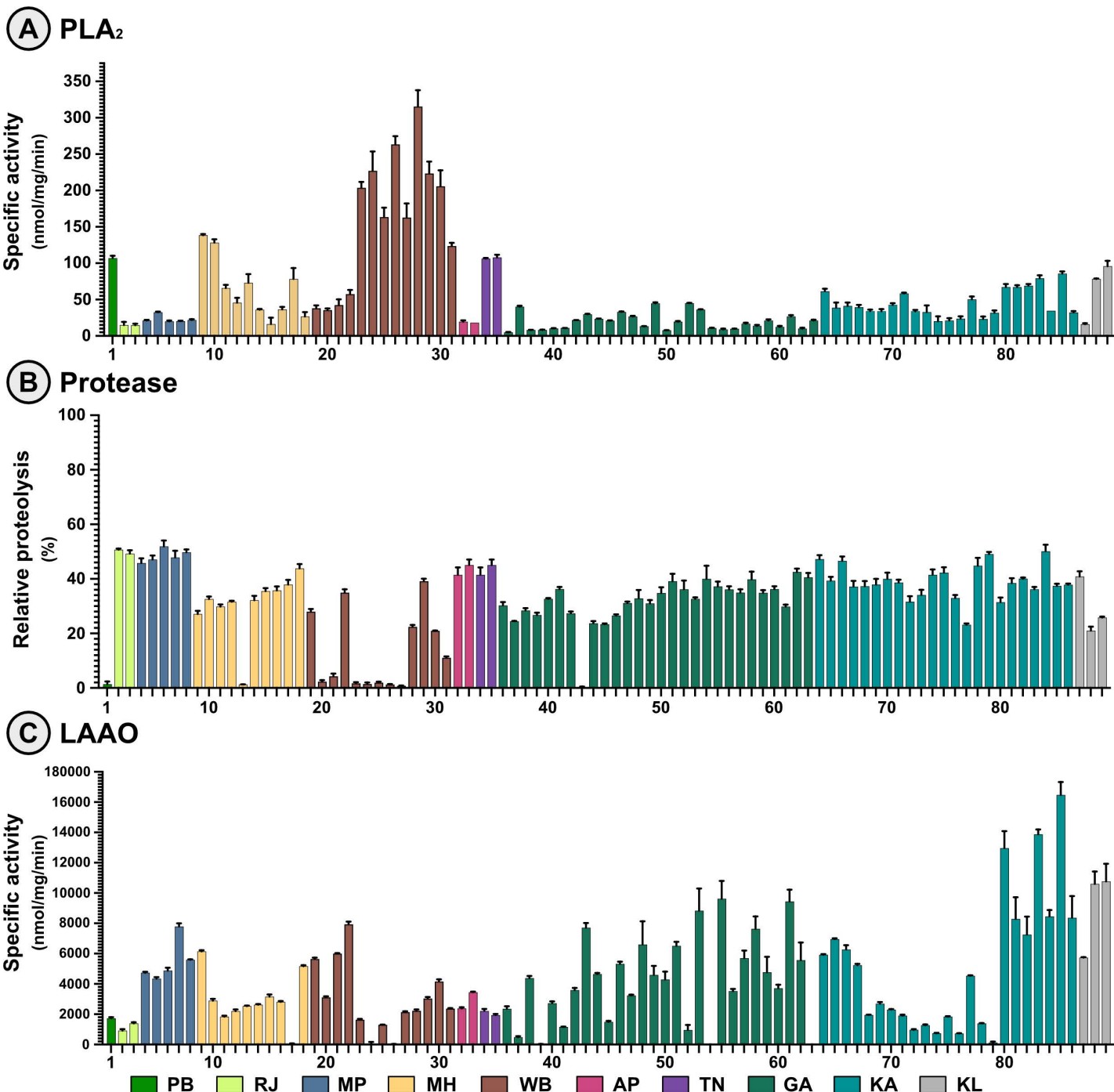

**Fig 2. Enzymatic activities of Russell's viper venoms.** Bar graphs depicting the (A) $PLA_2$, (B) protease and (C) LAAO activities of *D. russelii* venoms from across India are shown, where the x-axis represents sample IDs in Table A in S1 Text, while the y-axis represents specific activities (nmol/mg/min). The assays were performed in triplicates, and the mean activities were plotted, with the error bars indicating the standard deviation. Samples from each region/state are uniquely colour-coded. See Table A in S1 Text for details. Here, PB: Punjab (north India); RJ: Rajasthan (west India); MP: Madhya Pradesh (central India); MH: Maharashtra (southwest India); WB: West Bengal (east India); AP: Andhra Pradesh (southeast India); TN: Tamil Nadu (south India); GA: Goa (southwest India); KA: Karnataka (south India) and KL: Kerala (south India).

**Table 1. Bioclimatic predictors and their mathematical definitions.** This table depicts the downselected bioclimatic predictors with their mathematical representations. The table also provides the temporal and spatial scales and the predictors' biological relevance.

| Bioclimatic variable | Equation | Temporal: Spatial scale | Biological relevance |
|---|---|---|---|
| Annual mean temperature (AMT) | $\dfrac{\sum_{i=1}^{i=12} Tavg_i}{12}$ | Annual, global | |
| Temperature Annual Range (TAR) | $\max(Tmax_1,\ldots,Tmax_{12}) - \min(Tmin_1,\ldots,Tmin_{12})$ | Annual, global | Effects of ranges of extreme temperatures |
| Annual Mean Diurnal Temperature Range (AMDTR) | $\dfrac{\sum_{i=1}^{i=12}(Tmax_i - Tmin_i)}{12}$ | Monthly, local | Effects of Monthly Temperature Fluctuation |
| Isothermality (I) | $\dfrac{MDR}{TAR} \times 100$ | Comparison | Comparative influence of large to small-scale weather patterns |
| Temperature Seasonality (TS) | $100 \times \dfrac{SD(Tavg_1,\ldots, Tavg_{12})}{AMT + 273.15}$ | Monthly, local | Effect of local temperature variation |
| Annual Precipitation (APN) | $\sum_{i=1}^{i=12} PPT_i$ | Annual, global | |
| Precipitation Seasonality (PS) | $100 \times \dfrac{SD\left(Pavg_1,\ldots Pavg_{12}\right)}{1 + \dfrac{APN}{12}}$ | Monthly, local | Effect of Local Precipitation Variation |

dataset but only accounts for 17.8% of the variance in $PLA_2$ activity, as indicated by the adjusted $R^2$.

The SLR models developed to understand the role of bioclimatic factors on the venom proteolytic activity of the pan-Indian Russell's vipers revealed that some models were associated with relatively higher $R^2$ values. This suggests that abiotic factors could potentially influence the proteolytic activity of *D. russelii* venoms. However, they were concurrently marked by violations of the normality assumption, rendering the model predictions questionable. The breach of normality suggests that the current set of independent variables might not fully capture the underlying distribution of proteolytic activity or that the SLR model is not the most appropriate methodology to interpret such data.

Next, an SLR model was developed to explain the role of bioclimatic factors in driving the variation in venom LAAO activity of Russell's viper. However, these models did not yield significant predictions when individual bioclimatic variables were considered. Despite this, the model incorporating Temperature Seasonality (TS) as a predictor suggests a marginal influence of this variable on LAAO activity. The parameter estimator indicates an increase of 2.799 units in enzyme activity for every unit increase in TS. However, it only accounts for a mere 1.5% of the observed variability in LAAO activity, as denoted by the adjusted $R^2$ ($p > 0.05$). Further, the tests for homoscedasticity, linearity, and normality were satisfied in the models involving Annual Mean Diurnal Temperature Range (AMDTR) and TS, suggesting adequate model fit.

### 3.3 Multiple linear regression models

Through multiple linear regression analyses, we explored the influence of various interacting bioclimatic variables on the enzymatic activities of $PLA_2$, proteolytic enzymes, and LAAO

in Russell's viper venoms (Fig 3). The variability in PLA$_2$ activity was most comprehensively described by a model including TAR, AMDTR, and Annual Precipitation (APN), accounting for up to 22.64% of the variance ($R^2 = 0.2264$) with a highly significant p-value (p<0.001). This model passed all diagnostic tests except multicollinearity, indicating reliable predictive capacity without violations of regression assumptions. In contrast, proteolytic activity demonstrated a stronger relationship with the bioclimatic predictors, including TAR, AMDTR, Isothermality (I), TS, PS, and APN, with the best model explaining nearly 49.26% of its variance ($R^2 = 0.4926$) with high significance (p<0.001). MLR modelling of LAAO activity revealed that the most significant model only explained 10.71% of the variance ($R^2 = 0.1071$, p<0.001). Additionally, the AIC values of the models have also been included to provide robust statistical support (Fig A in S1 Text).

## 3.4 Prediction mapping

Utilising step-wise multiple linear regression models, we have generated spatial distribution predictions for three key enzymatic activities of *D. russelii* venom, namely PLA$_2$, proteases, and LAAO. PLA$_2$ activity map provided a unique spatial pattern, with increased activity across India's eastern and western coast (Fig 4A). Further, higher activities were also predicted in regions encompassed within the Gangetic plains, gradually dropping towards peninsular

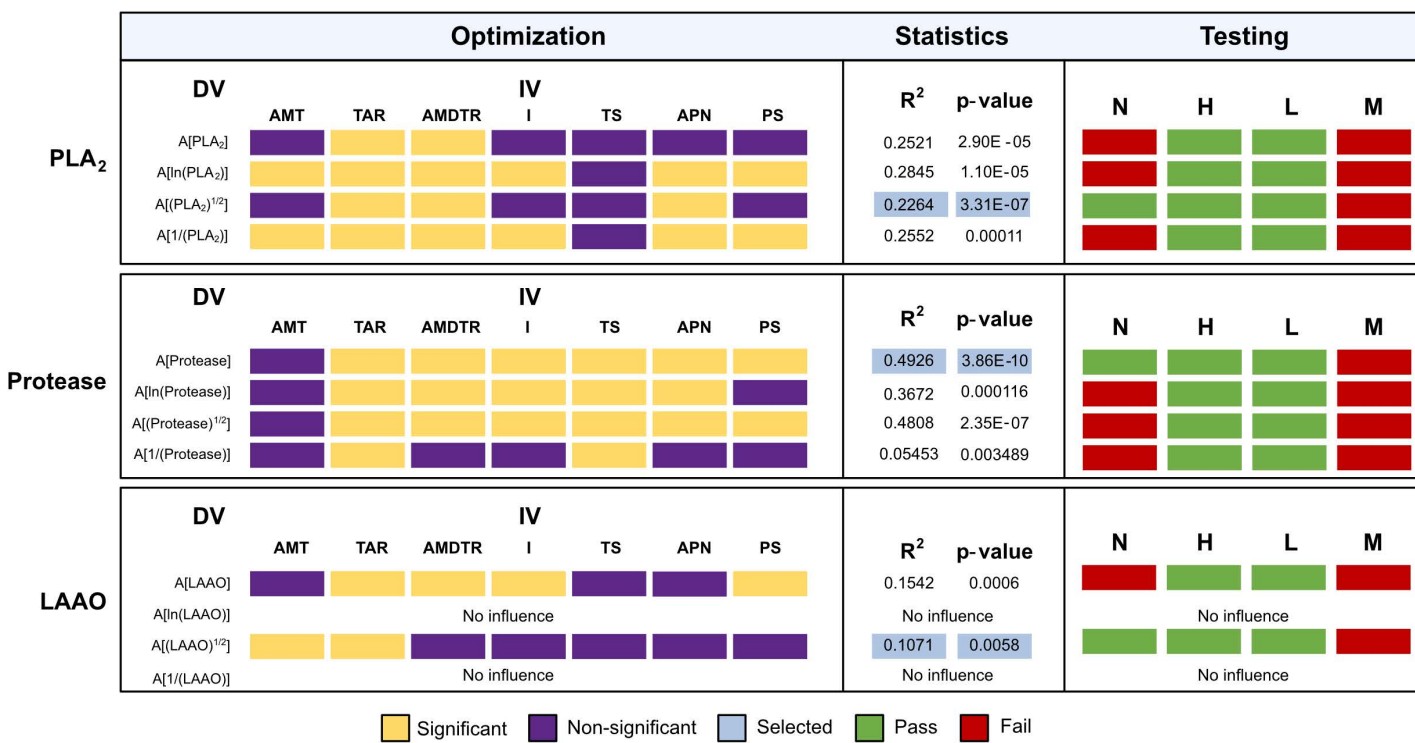

**Fig 3. Statistics of MLR models.** Various MLR models were built to explain the variation in the *D. russelii* venom PLA$_2$ protease and LAAO activities using the prevailing bioclimatic factors at the sampling locations. A combination of independent variables (IV), as depicted by yellow squares, was found to significantly affect the venom activities (A[enzyme]). The purple squares indicate variables that did not contribute substantially to the model. Logarithmic [ln(activity)], square-root [(activity)½], and inverse [1/(activity)] transformations of the dependent variables (DV) were also performed. The blue squares highlight the models that were downselected. Various model tests were also performed to assess normality (N), homoscedasticity (H), Linearity (L) and Multicollinearity (M) of the models. The green and red squares highlight if a particular MLR model passed or failed the corresponding model tests, respectively. The details of the bioclimatic variables included in the models are provided in Table 1. AMT: Annual mean temperature, TAR: Temperature annual range, AMDTR: Annual mean diurnal temperature range, I: Isothermality, TS: Temperature seasonality, APN: Annual precipitation and PS: Precipitation seasonality. The AIC values of the models are provided in Fig A in S1 Text.

India. On the contrary, the spatial distribution map constructed from the protease equation indicated pronounced regional variability, with the highest activity in the northwestern regions (Fig 4B). The proteolytic activity was predicted to decrease along the western coasts of India. In contrast, the LAAO activity appears to be the most uniformly distributed across the country, albeit with some hotspots of heightened activity (Fig 4C). This could also result from the MLR model capturing only 10% variation in the dataset.

## 4. Discussion

### 4.1 The combinatorial effect of bioclimatic factors on venom phenotypes

Evolutionary selection processes often shape phenotypic traits that vary significantly between populations, bestowing adaptive advantages to the organism in a particular niche [44,45]. Snake venoms are one of the befitting examples of such adaptive traits showing high variability at intra- and interpopulation levels in terms of composition, function, specificity, and potency [2,19,46]. However, the ecological and environmental interactions that drive phenotypic variation in venoms across populations are poorly understood [47]. Studies have attributed venom variation to three major determinants, namely, the biotic, abiotic and genetic components, but the relative importance of one over the other has not been established [5–8]. In addition to interspecific and conspecific competition, the prey and predators in an ecosystem are the major biotic components that influence venom composition. Particularly, mounting evidence has highlighted the effect of diet on venom complexity [10,48–50]. Similarly, venom resistance and other behavioural and physiological defence mechanisms in predators are also known to drive venom evolution to a certain extent [51–54]. At a population level, venom variation is documented in both large populations with high gene flow [11]

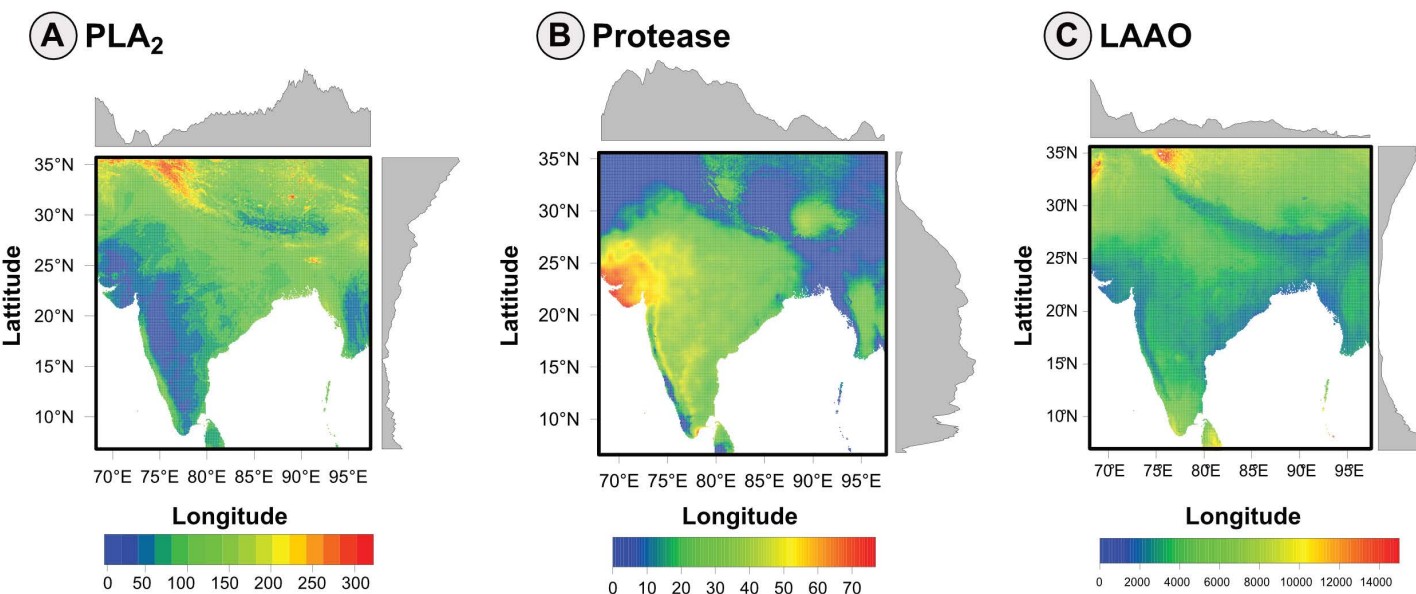

**Fig 4. Predictive mapping of functional venom variation in _D. russelii_ across the Indian subcontinent.** The predictive maps of _D. russelii_ venom (A) PLA$_2$ (B) protease and (C) LAAO activities based on the bioclimatic factors of the region are depicted on a map of India. The downselected MLR models, which establish the correlation between the bioclimatic variables and venom activities, have been used to build these predictive maps. The predicted enzymatic activity values across India are depicted by a colour gradient of blue to red, representing low to high activities. The grey graphs along the X- and Y-axis represent the activity at the individual pixels of 1 km$^2$ resolution. The maps were generated using the R packages terra and rasterVis. These maps directly visualize raster data sourced from WorldClim (http://www.worldclim.org, accessed on 14.04.2023), clipped to the geographic extent of India (approx. 68.1°E to 97.4°E longitude, 6.75°N to 35.7°N latitude). No external basemap or proprietary geographic layer was used.

and isolated populations with low or no gene flow [6,55,56]. In large connected populations, the local environmental conditions shape venom divergence patterns [11], while genetic divergence underpins venom variation in isolated populations. The cascading role of abiotic conditions on biotic factors was observed to shape venom phenotypes, emphasising the effect of local selective pressures on venom variability [12]. These studies have been predominantly conducted on *Crotalus spp.* with binary neurotoxic or haemotoxic phenotypes governed by two major snake venom toxins. Unfortunately, there has been a lack of focus on unravelling the effect of ecological and environmental drivers on venom variability in Indian snake species, owing to the complexity of their venom phenotypes. For instance, the cytotoxic and haemotoxic phenotypes of *D. russelii* venoms are determined by the relative abundances of major and minor toxin families, including SVMPs, serine proteases, PLA$_2$s, LAAO, lectins and vascular endothelial growth factors among many others [16–19]. There have also been reports of neurotoxic manifestations associated with Russell's viper bites [57,58]. Therefore, precisely modelling correlations between environmental components and venom phenotypes in *D. russelii* is complicated. In addition, forest department-enforced legal constraints are associated with collecting venom samples in India [59]. Despite these limitations, in this study, we collect over 115 *D. russelii* venom samples from 34 unique point locations across the Indian geography and attempt to decipher the complex interactions between the bioclimatic variables and the functional variation in venoms.

The results of SLR analysis in this study showed that temperature or precipitation alone does not significantly explain the observed variations in snake venom functions (Table B in S1 Text). Instead, multiple abiotic factors collectively influence the documented venom variation. Significant statistical models were obtained through MLR analysis, highlighting that a combination of variables, including global and local fluctuations in temperature and precipitation, collectively affect venom variability. Interestingly, the variations in activities of a major venom component, namely, the proteases ($R^2$ = 0.49, p<0.0001), were influenced by bioclimatic factors. In contrast, abiotic predictors did not affect the variations in a minor component, such as LAAO ($R^2$ = 0.10, p=0.0005). On the other hand, the linear regression models predicted the variability in PLA$_2$ activity only to a certain extent ($R^2$ = 0.22, p<0.0001). This could be associated with the selective enrichment of this toxin in certain populations, and the variability could be driven by other biotic and abiotic factors that were not considered in these models. Similar observations were recorded for the correlation between environmental variables and enzymatic activities of *C. viridis* venoms. The prediction of variability in the proteolytic activity was significant, whereas the PLA$_2$ activities did not have a significant relationship with the abiotic factors [12]. Therefore, these results reiterate that the functional venom variations observed in pan-Indian populations of *D. russelii* may be dictated by a complex interaction between a multitude of environmental determinants.

## 4.2  Predictive mapping of functional variation in *D. russelii* venoms

Based on the interesting insights from the regression analyses, the MLR models were used to build predictive maps to visualise the functional characteristics of *D. russelii* venoms across India. These maps extrapolate the MLR models to predict the enzymatic activities of *D. russelii* venoms in various regions directly based on local climatic data. The predictive maps revealed interesting functional diversity patterns across the Indian geography (Fig 4B). The model predicted that the proteolytic activity of *D. russelii* venoms would be high in northwestern India. Further, the proteolytic activity was observed to decrease as we progressed southwards along the coast of western India. The predicted proteolytic activities dropped to a minimal value (0-10%) in the eastern and northeastern parts of the Indian subcontinent. Overall, the model predicts increasing trends of proteolytic activity in drier regions. India's climatic

regimes could explain some of these observed patterns. Low precipitation levels characterise the arid and semi-arid regions of northwestern India. On the other hand, the evergreen forests of Western Ghats and northeastern India record overall high precipitation. Additionally, the temperature fluctuations within a diurnal cycle and between seasons are maximal in the dry, desert regions and minimal along the coasts and inside the tropical forests. While the land-sea breeze patterns maintain the ambient temperature along the coasts, the temperatures could plummet to extremely hot and cold conditions in deserts. Based on these findings, we theorise that *D. russelii* venoms in arid, semi-arid and dry regions will be characterised by higher proteolytic activities, whereas in humid and high-rainfall regions, they will likely exhibit lower proteolysis.

The predictive patterns observed for $PLA_2$ activity were in complete contrast to those observed for proteolytic activity (Fig 4A). The $PLA_2$ activity in peninsular and northwestern India was predicted to be low compared to the western and eastern coasts. Interestingly, high $PLA_2$ activity was predicted on the windward side of the Western Ghats (southwestern India), whereas the leeward side was predicted to have lower $PLA_2$ activity. Though only 22% of the observed variation in $PLA_2$ activities was explained by our model, the predictive map provides a broad overview of the prevalence of certain $PLA_2$-activity-rich zones across India. Finally, the prediction capability of the selected MLR models, which explained about 10% variation in the LAAO activity, was low (Fig 4C). The prediction mapping did not effectively capture the variations in this activity across India, further highlighting that the LAAO variations might be independent of the fluctuations in abiotic determinants. These maps collectively suggest a complex biogeographic pattern of venom variation likely driven by climatic conditions, which, in turn, may dictate prey diversity, predator pressure and/or competition. The predictive models have thus provided a valuable tool for visualising the potential distribution of these activities. They also shed light on the adaptive evolution of venom phenotypes across the diverse landscapes of the Indian subcontinent.

## 4.3  Potential applications of predictive modelling of venom phenotypes

Despite technological innovations and clinical advancements, snakebite incidents continue to pose a serious public health challenge in the Indian subcontinent. Among the medically important snakes of India, Russell's viper contributes to a significant proportion of snakebite-associated mortalities and immutable morbidities [14]. Though the mainstay treatment for Russell's viper bites is the administration of polyvalent antivenom, the effectiveness of the antivenom therapy is affected by the complexity of venom variability among individual snakes of this species across its distribution [19]. The venoms vary significantly in terms of toxin composition, abundance, and function across and within populations, thereby limiting the deployment of new therapeutic strategies. To address this problem, it is imperative to quantify and characterise the variation. However, quantifying the extent of variation through experimental analyses could be tedious due to the prevailing legal restrictions and logistical complications associated with collecting snake venoms. In this study, we demonstrate that by evaluating the bioclimatic predictors of a region and by leveraging statistical modelling, the functional characteristics of *D. russelii* venoms could be predicted. using the bioclimatic predictors in a region. These predictive maps will guide us in unravelling the geographic venom variation in a species and provide valuable insights into localities where a particular targeted therapy, such as a toxin-specific recombinant antibody or a species-specific antivenom or small molecule inhibitor drug, could be deployed. For example, snakebite cases in northwestern India require therapies targeting proteolytic components of *D. russelii* venom, while the lethal effects of *D. russelii* venoms in eastern India may require anti-$PLA_2$ therapeutics. While we acknowledge that venom toxins could inflict other physiological manifestations

besides the functional activities characterised in this study, these predictive maps may open avenues for further targeted experimental research. These predictive mapping strategies will be cost-effective and offer an ethical alternative to preclinical animal testing of snake venoms from distinct regions. These maps could also find potential applications in wildlife forensics, wherein functional activity testing of venoms could help trace the origin of the confiscated venom sample.

## 5. Limitations

This study analyses the influence of various bioclimatic factors on the phenotypic variations in the venoms of one of the medically most important viperid snakes from the Indian subcontinent. While our analyses highlight that the abiotic factors explain a significant proportion of the observed venom variation, the role of biotic factors such as diet and predator pressure may not be negated. Moreover, since this study is designed to decipher the influence of bioclimatic variables on a particular species under the given sampling variables, the observations may not necessarily apply to other closely related species. Furthermore, owing to the predictive nature of the models and the multicollinearity of the predictors, the inferences have been derived with biological relevance in mind. Since climatic effects are holistic, no one variable could be adjudged to be a strong influencer of the venom variation. In certain locations, the venoms from multiple individuals within a short geographic distance (<50 km) have been pooled due to logistical constraints and limited venom yield. However, the bioclimatic predictors remained relatively the same across these locations, and hence, these samples were included in the dataset. In this study, we have also accounted for the influence of various ecological factors on venoms. For instance, Russell's viper has not been documented to exhibit intersexual venom variation [60]. While ontogenetic shifts in venom profiles have been documented in this species [60], we negated these effects by specifically sampling venoms from adult individuals.

## 6. Conclusion

In summary, our research demonstrates that a combination of temperature and precipitation factors collectively explain the variation observed in the venoms of Russell's vipers across India. In particular, these bioclimatic conditions influenced the proteolytic and $PLA_2$ activities, whereas the minor venom components, such as LAAO, were unaffected. By employing multiple regression models, we identified key environmental drivers of venom variability, and through the generation of phenotype maps, we predicted venom biochemistry across the range distribution of *D. russelii*. Our study provides valuable insights to advance our understanding of the evolutionary adaptations of snake venoms to distinct ecological niches and has practical applications in improving snakebite management. Targeted antivenom therapies informed by our findings could enhance treatment efficacy and patient clinical outcomes and help reduce the high morbidity and mortality associated with Russell's viper envenomations. Our research underscores the critical role of environmental factors in shaping venom diversity and highlights the potential for integrating ecological data into therapeutic intervention of snakebites.

## Supporting information

**S1 File. The R codes for data analyses. Fig A.** Statistics of MLR models with AIC. **Table A.** Sampling location and bioclimatic variables. **Table B.** Statistics of GWR and MLR (Global) Models. **Table C.** Statistics of SLR models.
(PDF)

## Acknowledgement

The authors thank Ajinkya Unawane, Prasad Gond, Gerard Martin, Romulus Whitaker and Ajay Kartik for helping with the venom collection. Eeshan Damle for initial biochemical characterisation of venoms. The authors are extremely grateful to all the different State Forest Departments and the MoEFCC Government of India for kindly granting the required permissions to carry out this study.

## Author contributions

**Conceptualization:** Navaneel Sarangi, R R Senji Laxme, Kartik Sunagar.

**Data curation:** Navaneel Sarangi, R R Senji Laxme.

**Formal analysis:** Navaneel Sarangi, R R Senji Laxme, Kartik Sunagar.

**Funding acquisition:** Kartik Sunagar.

**Investigation:** Navaneel Sarangi, R R Senji Laxme, Kartik Sunagar.

**Methodology:** Navaneel Sarangi, R R Senji Laxme.

**Project administration:** R R Senji Laxme, Kartik Sunagar.

**Resources:** Kartik Sunagar.

**Supervision:** R R Senji Laxme, Kartik Sunagar.

**Validation:** Navaneel Sarangi, R R Senji Laxme, Kartik Sunagar.

**Visualization:** Navaneel Sarangi, Kartik Sunagar.

**Writing – original draft:** Navaneel Sarangi, R R Senji Laxme, Kartik Sunagar.

**Writing – review & editing:** R R Senji Laxme, Kartik Sunagar.

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
