## [Decision Letter · Decision Letter 0]

23 Sep 2024

Dear Dr. Sunagar,

Thank you very much for submitting your manuscript "Significant Serpents: Predictive Modelling of Bioclimatic Venom Variation in Russell’s Viper" for consideration at PLOS Neglected Tropical Diseases. As with all papers reviewed by the journal, your manuscript was reviewed by members of the editorial board and by several independent reviewers. In light of the reviews (below this email), we would like to invite the resubmission of a significantly-revised version that takes into account the reviewers' comments. 

We cannot make any decision about publication until we have seen the revised manuscript and your response to the reviewers' comments. Your revised manuscript is also likely to be sent to reviewers for further evaluation.

Sincerely,

Marco Aurélio Sartim

Guest Editor

Wuelton Monteiro

Section Editor

Reviewer's Responses to Questions

**Key Review Criteria Required for Acceptance?**

**Methods**

-Are the objectives of the study clearly articulated with a clear testable hypothesis stated?

-Is the study design appropriate to address the stated objectives?

-Is the population clearly described and appropriate for the hypothesis being tested?

-Is the sample size sufficient to ensure adequate power to address the hypothesis being tested?

-Were correct statistical analysis used to support conclusions?

-Are there concerns about ethical or regulatory requirements being met?

Reviewer #1: The objectives are well articulated and the design is slightly adequate, but the population is neither clearly described neither sampled to a satisfactory extent so as to meet the study aims.

I don't believe the sample size is sufficient for the hypotheses tested.

Some of the statistical analyses were deficient, and this is acknowledged by the authors but this was not taken into account to fine tune the analyses, or to evaluate the results shown.

Reviewer #2: • There are areas with low records of data for this species. It would be interesting to check in which eco-regions of India do you have records and in which other you do not have records. Also, the experimental setup for measuring toxicological effects is hard, so you should put that trade-off between having more samples across the country and having more samples to get a representation of the total distribution of this snake in India. In addition, it would be interesting to add the populations that you are covering in venom sampling for this species.

• How can you be sure that pulled venoms don’t generate a bias in terms of intraspecific variation or onthogenic variation? Please add the information about the age/size of each snake used in the pull and how do you can be sure that onthogenic variation does not affect your results. 

• Please improve the simple and multiple lineal regression analysis to correct geographically your results. How do you know that you don’t have spatial autocorrelation? If so, how do you ensure that your patterns are product of climate variables instead of spatial contiguity of locations? You can use geographically weighted regressions, or Bayesian inference in mixed effect lineal models by adding spatial autocorrelation as a random effect in the intercept, see ICAR and BYM models.

Reviewer #3: (No Response)

**Results**

-Does the analysis presented match the analysis plan?

-Are the results clearly and completely presented?

-Are the figures (Tables, Images) of sufficient quality for clarity?

Reviewer #1: The results do match the analyses plan

The results are clearly presented and figures are of sufficient quality.

Reviewer #2: • The x-axis labels in the figure 2 are not clear. Is it a variable? Are you presenting histograms for each region? Why do you have a scale from 1 to >80 there? Please add the label of x-axis if it is a variable, or remove labels and thicks if it is only the number of the sample that you are plotting.

• There is a mistake in table 1, in equation for PS. I can only see squares instead of the variables that are used in the expression. Please change the name of the last column because significance can be confounded with a statistical analysis that shows that this variable modifies your toxicological variables significantly.

• It is not clear the difference between your SLR and MLR. Why are you saying that you used SLR but you talk about the effects of multiple climatic variables? I understand that you are explaining each variable independly in different models, but it is only clear after looking at S2 table. Please enhance the way as you describe this section and also in methods: You are conducting different SLR for each independent variables and each SLR is independent of others SLR. In MLR it is not clear if you used also interactions between your variables. As a consequence, you are showing a lot of information of several models that are not good explaining your data, thus it is difficult to understand which model are you selecting as your best model to generate predictive maps. If you use GWR you can compare between all the possible combinations of the independent variables (Also evaluating each variable by itself) in different models and compare them by AIC or BIC, which punish the predictive capacity of the model by the number of parameters. As a consequence, you can select the model that explain best your data without doing overfitting by having a lot of parameters, and the use this model to generate your spatial predictions.

• The figure with the extrapoaltions done using your best model is confousing because the colorbar have above Longitude. Please include the label and the units for this colorbar, which I think should be the units of your dependent variables.

Reviewer #3: (No Response)

**Conclusions**

-Are the conclusions supported by the data presented?

-Are the limitations of analysis clearly described?

-Do the authors discuss how these data can be helpful to advance our understanding of the topic under study?

-Is public health relevance addressed?

Reviewer #1: I do not think that the analyses and results support the conclusions, mainly for two problems: The decision to not model spatial clustering inflates parameter estimates and p-values and the decision to not use generalised linear models do not allowed the authors to evaluate significance with the appropriate distribution of errors.

The relevance of the data are discussed and is relevant to public health, but as mentioned I do not think that the results as shown are reliable.

Reviewer #2: Discussion: 

• Your first paragraph is large and you are talking about different topics. I liked a lot the analysis of the neglection of environmental factors driving phenotypic characteristics of venom, which is crucial, but then you talk about the representativity of your data. Split this paragraph in two paragraphs.

• When you talk about the number of venom samples that you got, please refer to what I told you in methodology: You can have a lot of samples, but spatially they could be not so representative (i.e. if you are missing samples on different ecosystems/eco-regions/life zones). Please enhance the methodology by adding this so you can discuss of the real representation of your data over India.

• Please include about why did you not use anthropogenic intervention such as human footprint in your analysis. Can snakes sharing areas with humans during several years have different phenotypes in their venom?

• What about climate change? If climate is moduling venom phenotipic, what can we expect with the ongoing climate change and it effects in India? Can venom became more lethal? please add a paragraph discussing about climate change.

• I found your discussion well documented, and you touched really interesting points. I think that you just need more robustness in the methods to support strongly your discussion. Congrats it is a nice job!

Reviewer #3: (No Response)

**Editorial and Data Presentation Modifications?**

Reviewer #1: I recommend major revision.

Reviewer #2: (No Response)

Reviewer #3: (No Response)

**Summary and General Comments**

Reviewer #1: The subject of this article is highly relevant, and will provide useful information for public health authorities, medical practitioners and ecologists alike. My expertise falls within the spatial analysis side of this submission and this is what I have evaluated. A comprehensive review I think will need to include the opinion of molecular biology experts to.

“Parameter selection and data extraction”

I believe that referring to climatic data used as independent variables or covariates as “Parameters” is inadequate.

It s mentioned that bioclimatic data represent climate from 1895-2009. While adequate, we should be given additional information regarding the following aspects:

1. Did you use the bioclimatic variables calculated with the average of temperature and precipitation of the entire period? 

2. Or did you use the climatic conditions estimated for each year by matching the date the venom samples were collected?

3. If such a long term average climatologies were used, there should be a justification , as you later suggest that you avoided the variables that represent seasonality. Including these variables may be important even if you do not focus on spatio-temporal patterns because temporal variability is inherent to biological processes.

4. To extract the raster values using locations of venom samples, were these locations simple points or do they correspond to polygons?

“Linear regression models”

How exactly was enzymatic activity measured and so what are the units analysed? This is critical for the reader to understand the meaning of the models. Apologies if this was explained in detail previously, but given the amount of technical terms I could not identify this information in a straightforward way, so it would be useful to describe succinctly in this section.

At this point we should be given an idea of where were the samples obtained from, to assess how complete is the spatial coverage of the samples. This should be as complete as possible, because the use you are looking for these analyses is spatial prediction.

I also believe that, given the spatial nature off your analyses you should have used a spatially explicit analytical method to avoid pseudo-replication due to spatial autocorrelation.

Results

Ok, so the units of the data are mg/nmol/min, meaning that the units are min-1, which makes the data more suitable for a gamma model.

By inspection of the spatial distribution of the sampling points, there appear to be very large gaps and some point clusters. This would invalidate the analyses and the statistical inferences regarding the effect of the bioclimatic variables cannot be trusted unless point clustering is explicitly modelled.

I agree that the predictions are very questionable. Some of the problems may be fixed if a more adequate type of model is used such as gamma, given the units of the data and spatial clustering. Accounting for spatial clustering tends to reduce the size of estimated coefficients and increase p-values.

MLR models

A further test would be to analyse the spatial correlation of model residuals.

Predictive mapping

I don’t think neither the spatial coverage of the data nor the methods make for models suitable for the intended use. One work around would be to show that the resulting maps can roughly predict enzymatic activity, using cross validation, for instance selecting data points as independent as possible. Alternatively, mapping may be restricted to data used and show interpolated values rather than using covariates with very weak effects on the response variable and potentially with extrapolation issues after projecting to the entire geographic extent of India.

Reviewer #2: The articles looks for a interesting question that is how does climate drive phenotypic variation of venom activity. This is really interesting in the scope of climate change, and also understanding the ecological driving forces that modulate venom plasticity in venomous snakes species. The article is well written, and discussion is clever. 

There are points that must be enhanced, principally in data analysis and robustness of spatial methods, but the relevance of this article in toxinology and snakebite envenoming is high. This article will be worthy to be published in the journal after enhancing the robustness of data analysis.

Reviewer #3: The manuscript „Significant Serpents: Predictive Modelling of Bioclimatic Venom Variation in Russell’s Viper“ by Prof. Sunagar and colleagues presents a beautiful synthesis of functional venom profiling and bioclimatic modelling using a medically very important venomous snake species and a global hotspot of snakebite as models.

The authors use their innovative setup to show that several abiotic factors may be explanatory for venom variation (on functional level), a perspective so far seldomly investigated. The paper is well-written and addresses both, its novelty and limitations sufficiently. The study is methodologically sound and with the introduction of climatic modelling into functional venomics, the team of Prof. Sunagar is stepping onto new avenues for snakebite research. Thanks to this, the paper may have long-term impact by creating intangible assets (in form of venom phenotype maps that can guide first aid and medical care). I particularly liked the functional angle, since most works in the field largely rely on venom compositions derived from proteomics. However, the inclusion of functional assays targeting key viper venom activities paired with the extraordinary high sample size and geographic coverage, allow the authors to dive deep into the particularities of their data.

While reading through the presented manuscript, I really only could identify minuscule issues to point out, which were fully semantical in nature and thus would be purely based on personal preference. Hence, I decided to not nitpick with the team and not suggest further changes. I believe this is an excellent study and I hope that future works are carried out similarly on other important snake species.

PLOS authors have the option to publish the peer review history of their article (what does this mean? ). If published, this will include your full peer review and any attached files.

**Do you want your identity to be public for this peer review?** For information about this choice, including consent withdrawal, please see our Privacy Policy .

Reviewer #1: No

Reviewer #2: Yes: Carlos Bravo-Vega

Reviewer #3: No
---

## [Decision Letter · Decision Letter 1]

3 Feb 2025

PNTD-D-24-01066R1Significant Serpents: Predictive Modelling of Bioclimatic Venom Variation in Russell’s ViperPLOS Neglected Tropical Diseases Dear Dr. Sunagar, Thank you for submitting your manuscript to PLOS Neglected Tropical Diseases. After careful consideration, we feel that it has merit but does not fully meet PLOS Neglected Tropical Diseases's publication criteria as it currently stands. Therefore, we invite you to submit a revised version of the manuscript that addresses the points raised during the review process. Please submit your revised manuscript within 30 days Mar 04 2025 11:59PM. If you will need more time than this to complete your revisions, please reply to this message or contact the journal office at plosntds@plos.org. Please include the following items when submitting your revised manuscript: * A rebuttal letter that responds to each point raised by the editor and reviewer(s). You should upload this letter as a separate file labeled 'Response to Reviewers '. This file does not need to include responses to any formatting updates and technical items listed in the 'Journal Requirements' section below.* A marked-up copy of your manuscript that highlights changes made to the original version. You should upload this as a separate file labeled 'Revised Manuscript with Track Changes '.* An unmarked version of your revised paper without tracked changes. You should upload this as a separate file labeled 'Manuscript '. If you would like to make changes to your financial disclosure, competing interests statement, or data availability statement, please make these updates within the submission form at the time of resubmission. Guidelines for resubmitting your figure files are available below the reviewer comments at the end of this letter. We look forward to receiving your revised manuscript. Kind regards, Wuelton MonteiroSection EditorPLOS Neglected Tropical Diseases

Shaden Kamhawi

co-Editor-in-Chief

Paul Brindley

co-Editor-in-Chief

**Journal Requirements:**

1) Some material included in your submission may be copyrighted. According to PLOSu2019s copyright policy, authors who use figures or other material (e.g., graphics, clipart, maps) from another author or copyright holder must demonstrate or obtain permission to publish this material under the Creative Commons Attribution 4.0 International (CC BY 4.0) License used by PLOS journals. Please closely review the details of PLOSu2019s copyright requirements here: PLOS Licenses and Copyright. If you need to request permissions from a copyright holder, you may use PLOS's Copyright Content Permission form.

Potential Copyright Issues:

i) Figure 1. Please confirm whether you drew the images / clip-art within the figure panels by hand. If you did not draw the images, please provide (a) a link to the source of the images or icons and their license / terms of use; or (b) written permission from the copyright holder to publish the images or icons under our CC BY 4.0 license. Alternatively, you may replace the images with open source alternatives. See these open source resources you may use to replace images / clip-art:

ii) Figure 1 :Thank you for stating that "The map of India shown here was prepared with QGIS 3.8. Please (a) provide a direct link to the base layer of the map (i.e., the country or region border shape) and ensure this is also included in the figure legend; and (b) provide a link to the terms of use / license information for the base layer image or shapefile. We cannot publish proprietary or copyrighted maps (e.g. Google Maps, Mapquest) and the terms of use for your map base layer must be compatible with our CC BY 4.0 license.

iii) Figure 4:Thank you for stating that "The maps were generated using the R packages terra and rasterVis. These maps directly visualize raster data sourced from WorldClim (http://www.worldclim.org)”. The data on WorldClim is for non-commercial use only which is not compatible with our CC BY 4.0 license (which permits commercial use).  Please amend the figure so that the base map used is from an openly available source. Alternatively, please provide explicit written permission from the copyright holder granting you the right to publish the material under our CC BY 4.0 license.

2) Please amend your detailed Financial Disclosure statement. This is published with the article. It must therefore be completed in full sentences and contain the exact wording you wish to be published.

**Reviewers' comments:** Reviewer's Responses to Questions

**Key Review Criteria Required for Acceptance?**

**Methods**

-Are the objectives of the study clearly articulated with a clear testable hypothesis stated?

-Is the study design appropriate to address the stated objectives?

-Is the population clearly described and appropriate for the hypothesis being tested?

-Is the sample size sufficient to ensure adequate power to address the hypothesis being tested?

-Were correct statistical analysis used to support conclusions?

-Are there concerns about ethical or regulatory requirements being met?

Reviewer #2: 1. Please include explicitely which samples were pooled and which were not pooled. Include this as different colours in the dots presented in Figure 1.

2. In PLA2, Protease and LAOO activities assay section, you are using the same quantity of mass of different venom and diluting it in the same quantity of NOB substrate. However, you said that you measured protein concentration per venom. I would recommend to clarify how can you do the analysis without including the possible differences in venom concentrations per each sample, which can cause different ammount of enzimatic activites instead of a variation in composition.

3. It is not clear why did you removed seasonal data such as Max temperature at warmer month and the others. This data only uses the warmest month without assuming that that max temperature occured in the same month each year, of that the warmest mont was the same each year, which can be seasonal. Also, even if these variables talk about seasonality they are really important in the ecology of snakes: There are different ecological traits between rattlesnakes in U.S. (Seasonal temperature) and tropical rattlesnakes (Quite "constant" temperature). Although you don't have data for same snake per each time in the year (Which can be another great idea for another work to seek how "plastic" can venom expression be), a spatial area with high seasonality and another one without that seasonality can be really different and shape different condition. I recommend to include these variables in the models (In my experience, these variables shape really well difference of distribution between Bothrops vipers, because they define different ecological areas).

4. The statistical lineal models are well explained, and I find them really simple to understand which is really nice because you can explain the variation with basic models thus supporting really well your hypothesis. The selecion of GWR is also really nice for accounting for spatial autocorrelation. I found these sections nice.

**Results**

-Does the analysis presented match the analysis plan?

-Are the results clearly and completely presented?

-Are the figures (Tables, Images) of sufficient quality for clarity?

Reviewer #2: 1. Figure 1 and Figure 2: I liked both figures, but in figure 2 is kind of difficult to assign for each spatial label the spatial area in the map. Please include in Figure 2 each one of the areas that you refer in Figure 2 (PB, RJ, etc) as a vector so readers can locate easily in the map each area. Also, this can help to readers to determine which areas can be undersampled.

2. As I told you in methodology, several variation between activities could be caused by different venom concentrations, which you measured but didn't use in the models. Thus, you cannot attribute your found differences to composition because they can be caused by concentrations. Maybe a way for looking at this without including concentration in your models is to perform correlation matrix analysis (Pearson or Spearman) between your enzimatic activities: You compute the correlation between all the samples for PLA2 activity with LAAO, then between PLA2 and Protease, and finally between LAAO and protease. With this analysis, if you find a strong correlation between activities it tells you that some samples have strong activities for all kind of enzimatic activites, whilst others have low activites but also for all, which can be telling you that venom concentration is the mechanism that is causing the differences. Another way for doing this is including in your lineal models (Indepently and also in the multiple lineal models) venom concentration.

3. Please include explicitely that GWR results didn't show spatial autocorrelation, and that thanks to that you only used lineal models without any kind of spatial structure.

4. The final step of performing these risk maps based on enzimatic activities is really clever! I think that including and correcting these possible bias, you can generate risk maps with more robustness and also understand deeper the mechanism behind this variation. This topic is fascinating, and reading and looking these results is really nice. I find very useful to include ecology to understand toxinological variation, it is a novel topic so I want to congrat you for this work.

**Conclusions**

-Are the conclusions supported by the data presented?

-Are the limitations of analysis clearly described?

-Do the authors discuss how these data can be helpful to advance our understanding of the topic under study?

-Is public health relevance addressed?

Reviewer #2: 1. You attribute the changes in enzymatic activites with venom conposition, but given that you are not testing if differences are caused by different protein concentration in venoms you cannot conclude this. Please include concentration as told you before, so you can determine if differences are caused protein concentration (Which can be fascinating because it will imply a different way of variation by saving resources by producing venoms with not so much proteins, which are expensive. Thus, maybe for these populations evolution shapes that selecting forces makes for snakes more important to invest energy in other stuff rather than producing a high concentrated venom with higher enzymatic activities), or if they may be changed by venom composition (Which is another way that can be more related with different genetic expression).

2. Please include a paragraph about the limitations of pooling venoms for these toxicological studies: You are addresing this issue as using most of the venoms at an individual level, but this is no acknowledged in the discussion. Please aslo explain explicitely that you avoided onthogenic shift by using only adults, but that if reproduction data is inexistent for ensuring that all individuals are adults based on body measurements such as SVL pooling must be done carefully. Also relate here to possible intersexual differences in venom composition and add the references.

3. I would love to see a second part of the paper performing proteomic analyses between population to identify possible variations in protein composition, by including something that I found sometimes neglected in toxinology: The individual variation of venom composition. You had some pooled venoms which can have bias, but in general you are trying to go at an individual level. Several toxinologic studies pool venom, so you can get imformation about the averaged values of venom variables, but you don't get information about this dispersion: You cannot test if the found differences are significant.

**Editorial and Data Presentation Modifications?**

Reviewer #2: I find english really well written and easy to understand. Also figures are good and explicative. Please go to each section comments to see which revisions should be done.

**Summary and General Comments**

Reviewer #2: This work account for a novel and fascinating topic: Looking at geographic variation of venon toxiconology by including statistics, thus having statistical tests to account for statistical significance in that variation. Several toxinologic studies uses pooled venoms and only refer to mean values to tell about differences, but this neglecet the intrinsic variation between venom in the comparet populations, so they cannot assure that the differences exist and are not caused by randomness.

The study have one limitation that can be addressed to generate a great work: 1. Measuring protein concentration of venom but not using it in the lineal models to account if the activities differences are caused by protein concentrations rather that venom composition

I suggest to correct this issues, or to account and aknowledge these important limitations in the paper, to generate a comprehensive study that address an important topic: ¿Why is venom variating between regions?

As a personal comment, I didn't like the title "Significant serpents". I think that this beginning is not related and is not highlighting the real importance of your work.

PLOS authors have the option to publish the peer review history of their article (what does this mean? ). If published, this will include your full peer review and any attached files.

**Do you want your identity to be public for this peer review?** For information about this choice, including consent withdrawal, please see our Privacy Policy .

Reviewer #2: **Yes: ** Carlos Bravo-Vega

---

## [Editor Report · Decision Letter 2]

24 Feb 2025

Dear Dr. Sunagar,

We are pleased to inform you that your manuscript 'Significant Serpents: Predictive Modelling of Bioclimatic Venom Variation in Russell’s Viper' has been provisionally accepted for publication in PLOS Neglected Tropical Diseases.

Best regards,

Marco Aurélio Sartim

Guest Editor

Wuelton Monteiro

Section Editor

Shaden Kamhawi

co-Editor-in-Chief

Paul Brindley

co-Editor-in-Chief

---

## [Editor Report · Acceptance letter]

Dear Dr. Sunagar,

We are delighted to inform you that your manuscript, "Significant Serpents: Predictive Modelling of Bioclimatic Venom Variation in Russell’s Viper," has been formally accepted for publication in PLOS Neglected Tropical Diseases.

Best regards,

Shaden Kamhawi

co-Editor-in-Chief

Paul Brindley

co-Editor-in-Chief
